# Pion Detection Using Single Photon Avalanche Diodes

**DOI:** 10.3390/s23218759

**Published:** 2023-10-27

**Authors:** Anthony Frederick Bulling, Ian Underwood

**Affiliations:** School of Engineering, Scottish Microelectronic Centre, The University of Edinburgh, Edinburgh EH9 3FB, UK; ian.underwood@ed.ac.uk

**Keywords:** single photon avalanche diode, SPAD array, pion, CMOS image sensor, heavy charged particle detection, high-energy physics

## Abstract

We present the first reported use of a CMOS-compatible single photon avalanche diode (SPAD) array for the detection of high-energy charged particles, specifically pions, using the Super Proton Synchrotron at CERN, the European Organization for Nuclear Research. The results confirm the detection of incident high-energy pions at 120 GeV, minimally ionizing, which complements the variety of ionizing radiation that can be detected with CMOS SPADs.

## 1. Introduction

Previous research has demonstrated the capability of CMOS SPAD arrays in the detection of ionizing radiation [1,2,3]. Ionizing radiation, with or without mass, incident upon an SPAD depletion region produces electron–hole pairs, and coupled with Geiger operation, the induced electric field and large inherent gain, will generate impact ionization and subsequent avalanche breakdown, therefore resulting in the detection of incident ionizing radiation using SPADs [4].

The motivation for the investigation of SPAD technology in the detection of ionizing radiation, specifically heavily charged particles, is two-fold. First, SPAD technology has developed substantially over the past decades, and more significantly, integration into CMOS has provided a platform of highly integrated, miniaturized and cost-effective avalanche sensors, proficient at expeditious quantum detection [5,6].

Secondly, SPADs function in a manner akin to ionizing radiation detectors that use hybrid-silicon sensors [7,8]; it is believed that with SPAD advancement and maturity over the past two decades in CMOS, this parallel and rapidly developing technology may lend itself favorably towards progress and application in the detection of ionizing radiation. It is limited not only in high-energy particle physics, but every-day commercial and point-of-care applications, such as radiation imaging, beam therapy and electron microscopy, which would greatly reduce cost and help advance miniaturization of current technologies, improving both availability and mobility. However, the primary limitation of SPADs is the mechanism for detection, which simply registers an event, unlike high-energy physics pixel detectors, which can determine the total energy loss with incident ionizing radiation. SPADs operate well in environments where the type of radiation source is already known and abundant and the inherent dead time is not a limitation. Furthermore, one of the integral concerns with using SPADs for the detection of ionizing radiation is the potential radiation damage to the SPAD detector.

A consequence of their mature development in advanced processes, detection sensitivity, micron-sized pixels and timing capabilities, CMOS SPADs have found their way into numerous applications. The most notable are biomedical fluorescence lifetime imaging microscopy (FLIM), Light Detection and Ranging (LiDAR), Time-of-Flight (ToF) imaging, single photon counting (SPC), high-speed imaging, biological particle tracking and visible light communications [9,10,11,12,13,14,15,16]. However, all prominent applications are constrained to the visible and near infrared spectrums. The advances and broad applications of CMOS SPADs have led to increased R&D and financial backing as a commercially viable technology. The motivation is towards further miniaturization, performance improvements, 3D stacking and developing highly integrated intelligent sensors with abounding integral capabilities [17,18].

## 2. Three-Dimensional SPAD Image Sensor

The SPAD image sensor used for this research, hereinafter referred to as MINI3D, is a 120 × 128-pixel CMOS miniature time-resolved SPAD array, and the first presented 3D-stacked backside-illuminated (BSI) silicon sensor capable of both time-resolved imaging and single photon counting (SPC) [6,13]. Three-dimensional stacking technology is driven by the demand for increased transistor density and is a key technology in dynamic random access memory (DRAM) fabrication and high bandwidth memory interface (HBM) technology to improve both bus speeds with shorter bus paths, and reduce overall power consumption by reducing RC parasitics [19,20]. The pursuit of 3D-stacked CMOS SPAD technology is seen as a key process in realizing miniature CMOS SPADs with highly integrated circuitry capability, which would otherwise be unachievable in monolithic CMOS technology [6]. Three-dimensional stacking permits high-pixel array fill factors that are scalable, while optimizing pixel driver circuitry without compromise, by separating the sensor and integrated circuit tiers to deliver optimal performance and capability [6].

The MINI3D image sensor pixel comprises two tiers, connected with a wafer-to-wafer hybrid bonding connection, shown in Figure 1 [6,21]. The SPAD image sensors are implemented on the top tier, within an imaging 65 nm CMOS process, without active circuitry, for an optimized fill factor of 45% and pixel pitch of 7.83 µm [6]. The bottom tier houses all the necessary integrated circuitry and is implemented within a complimentary 40 nm process [6]. The stacked wafers are bonded at both the hybrid bond (HB) Cu-to-Cu and oxide interfaces using a dual damascene integration [6]. Each pixel has a dedicated 12-bit ripple counter providing a binning capacity for 4096 photon counts [6].

The MINI3D SPAD image sensor structure resembles a hybrid pixel detector used in the ATLAS experiment at the CERN large hadron collider (LHC), and is therefore expected to have improved radiation hardness capabilities [22].

The top tier is inverted, allowing incident photons to enter the backside of the die, referred to as a BSI SPAD image sensor, where the incident material is silicon. The backside of the die has been wafer-thinned and the thickness is undisclosed to authors. The SPAD junction is a P-well to shared retrograde Deep N-well with an isolated substrate and NMOS quench transistor [6]. 

An extensive overview of the MINI3D SPAD architecture, circuitry, performance characterization and comparison to other sensors is provided in [6]. For the external operation and interface with a computer, all operational, control and supply signals are externally generated and connected using a field-programmable gate array (FPGA).

## 3. Proton Synchrotron and Pion Generation

For proton–proton collisions at the LHC, the journey begins with hydrogen, which is ionized using an electric field to strip off electrons and produce bare protons [23]. The protons are separated into bunches and accelerated up to 50 MeV in linear accelerator 2 (Linac 2) and delivered to the Proton Synchrotron Booster accelerator. The Booster accelerates the protons to 1.4 GeV and injects the protons into the Proton Synchrotron (PS) [24], which further accelerates the protons to 26 GeV [25]. In addition to protons, the PS also accelerates a range of other particles, e.g., electrons and helium nuclei [25]. The accelerated particles from the PS are injected into the Super Proton Synchrotron (SPS), which are accelerated up to 450 GeV and finally injected into the LHC for final acceleration and ensuing collision, which create rare processes, allowing scientists to investigate particle properties and provide an improved understanding of matter [26]. 

To explore the detection of heavy charged particles using CMOS SPADs, the study conducted experiments at CERN’s Super Proton Synchrotron (SPS) facility, where hadrons are generated by colliding accelerated particles with fixed targets [27]. By colliding 400 GeV protons from the SPS into a fixed beryllium target plate [28], the experiment generates hadrons. These hadrons are subsequently directed towards the north area, where detectors measure their properties. A wobbling station is employed to separate specific hadrons into various beam lines, enabling researchers to conduct measurements on select hadron beams and experiment with new detector technologies [27]. The beam line available for testing was the pion beam line at CERN.

A pion, symbolized as π, is a hadron and is specifically classified as a meson. Mesons are composite particles consisting of one quark and one antiquark that are bound together with the strong force. Pions exist in three variations: positive (π^+^), negative (π^−^) and neutral (π^0^). The charge of a pion is determined with the combinations of up, anti-down, anti-up and down quarks it contains. Pions are the lightest of mesons and are unstable composite particles and therefore decay. Charged pions have a mean lifetime of 26.033 (±0.005) ns, whereas the neutral pions have a mean lifetime of 85.2 (±1.8) as. The masses of charged and neutral pions are approximately 139.5706 MeV/c^2^ and 134.9770 MeV/c^2^, respectively, which are roughly 273 times greater than the rest mass of an electron [29,30]. 

## 4. Experimental Setup and Parameters

For the operation of the SPAD image sensor, four distinct operational bias voltages were selected, namely 12.5, 13, 14 and 15 V. An increase in reverse bias voltage results in a proportional increase in detection probability, however, at the cost of increased spurious counts, quantified as dark count rate (DCR). Prior to each measurement using the SPAD image sensor, a control data set is captured at each bias voltage, from which the standard deviation and mean values for each pixel (15,360 for MINI3D) are calculated. With the mean and standard deviation of each pixel calculated, the mean and standard deviation of the sample means can be calculated. Thereafter, each pixel mean is compared to the mean of the sample mean and if the pixel mean does not lie within 2 standard deviations, a confidence interval of 95%, the pixel is flagged as a high dark count rate (HDCR) pixel. This allows for the identification and removal of HDCR pixels and will remove spurious counts during the analysis. Furthermore, for all readings from the SPAD image sensor, light blocking tape was placed over the sensor packaging to remove surrounding ambient light.

During pion irradiation, after HDCR pixels have been removed, and to ensure improved detection integrity during beam irradiation, a confidence interval of 4*σ* was selected. This was based on the known normal distribution DCR for each pixel and their calculated mean and standard deviation. Importantly, measurements were conducted under ambient temperature conditions. This, along with the chosen confidence interval, ensures that certain environmental factors like temperature were well controlled and did not introduce undue variability. Consequently, during beam irradiation for each frame, the pixel count is compared with the pixels’ control distribution. If the count value exceeds the threshold of 4*σ* from that respective pixel’s DCR mean and standard deviation, the value is retained. Otherwise, it is rejected to improve signal contrast. One of the limitations of the back-end read-out hardware was the delay in data output via the FPGA, as the firmware of the FPGA is fixed to 200 MHz. The delay between frame capture for a 1 ms frame exposure time was measured to be approximately 71 ms. If the frame exposure time is increased, this results in both increased DCR and increased data collection time, which results in reduced SNR and a proportional delay; therefore, a 1 ms frame exposure time was selected. The delay in subsequent frame captures poses an efficiency problem, and the particle accelerator delivers particles in spills, where each spill is made up of particle bunches with a predefined frequency and intensity. Therefore, for a 1 ms frame exposure time and subsequent 71 ms delay, the total time detection efficiency is approximately 1.4%.

For the detection of high energy pions, the H6 beam line at CERN’s North Area was used, which is a high-resolution secondary beam line. The primary 400 GeV proton beam from the SPS is directed onto a fixed beryllium target, which generates a secondary beam, a mixture of hadrons and electrons, where an incident lead absorber is used to remove electrons. The pion beam delivered to line H6 is based on priorities of other projects conducted at the time, including programs of all facilities served by the SPS and LHC. These factors contribute to the variability observed in the beam line spill cycle, spot size, intensity and beam energy. Nonetheless, during irradiation, the spill cycle lasted for approximately 5 s, the spot size resolution was approximately 5 × 5 mm^2^, the intensity was approximately 10^7^ particles per spill and the beam energy was 120 GeV [31,32]. 

Given the known time detection efficiency of 1.4%, the active SPAD can detect approximately 140,000 pions/spill. Furthermore, assuming an evenly distributed pion spill, the SPAD image sensor, with a 1 ms exposure time and a spill cycle of approximately 5 s, can capture roughly 69 frames/spill. Each frame would detect an estimated 2000 pions. However, the beam spot size has not yet been taken into account in these calculations. 

The spot size, as measured, has an approximate root mean square (rms) area of 25 mm^2^ [32]. In contrast, the active pixel area of the SPAD image sensor is 0.94 mm^2^ [6]. Consequently, the image sensor only covers 3.76% of the beam area. As a result, the SPAD image sensor is exposed to approximately 75 pions/frame (75 kHz) across its 15,360 pixels. Under ideal beam conditions, featuring an intensity of 10^8^ particles/spill, a spot size area of 12.6 mm^2^ and a minimum spill time of 4.8 s, the beam exposure increases to approximately 1554 pions per frame (1.55 MHz).

Managing the numerous settings of beam elements, like magnet currents and collimator openings, is intricate and requires the use of specific ‘beam files’ tailored for different beam types and user requirements. Different beam momenta; electron, hadron and muon beams; and varying wobbling configurations necessitate multiple files for a single beam line. These complexities highlight the challenges faced especially when one is not the primary user of the hadron detection setup [31].

The SPAD image sensor and printed circuit board with FPGA was fixed to the Malta plane detector telescope [33]. The SPAD image sensor location was aligned to a Malta detector plane and along with the 3D motor stage on the telescope (1 mm tolerance), the SPAD array was positioned within the beam line. A scintillation counter was used to trigger data capture during spill cycles and subsequently pause data capture to ensure continuous pion detection. Figure 2 provides a schematic diagram of the experimental setup. 

For probable detection, it is important to determine the average DCR noise floor of the SPAD image sensor including respective standard deviation for each pixel. The expected exposure intensity for the pion experiment across the entire CMOS SPAD array of 15,360 pixels is low, equating to approximately 4.88 Hz, which equates to 0.0488 pions/pixel per frame. For example, in [13], the measured average DCR for the MINI3D CMOS SPAD image sensor used at the SPAD bias voltage of 13 V is approximately 2 kHz for 1 ms sequential binning. The total dark counts across the entire image sensor are approximately 31 MHz. Therefore, as the DCR of the MINI3D SPAD image sensor is significantly larger than the predicted detections/frame as a result of relative low intensities for the pion beam, the detection is not discernible over the inherent DCR of the SPAD. Therefore, a significant data sample is required to reduce the standard error of the mean, as the sample means will cluster closer to the true population mean, allowing potential identification of the incident particles.

## 5. Experimental Results

For the pion experiment, there were a total of four irradiation sets of measurements taken, named E1, E2, E3 and E4, respectively. The total number of experiments was limited by beam availability and approved access. For each irradiation exposure, a control set of data was taken before irradiation, and HDCR pixels were filtered. A total of 43,500 frames of control data was taken at each SPAD bias voltage. For each irradiation, the experimental procedure was as follows:Beam access granted, access to beam line area H6 for setup;SPAD image sensor hardware installed and mounted to telescope;Remote access confirmed and initialization of SPAD image sensor;Control data set initiated for SPAD bias voltage of 12.5 V for predetermined number of frames, then SPAD bias voltage changed from 12.5 V to 13 V to 14 V to 15 V in turn, and respective control data sets taken;SPAD image sensor is reset and predetermined frames per irradiation cycle are set;SPAD frame capture initiated for each pion beam spill and paused after each spill. Run and pause were triggered using a scintillator;Once the predefined number of frames are captured, the SPAD bias voltage is changed from 12.5 V to 13 V to 14 V to 15 V in turn until the current cycle is complete. Then, a new cycle of measurements is taken and so forth, until the beam run ends;After irradiation, a control data set is taken. The SPAD image sensor is then powered off and hardware removed until beam access is granted once more.

Table 1 provides an overview of each experiment with total frames taken before, during and after irradiation including respective start and end dates in date month year format.

To adequately analyze the data and to determine the DCR noise floor of the SPAD image sensor, at each respective bias voltage, the average DCR for each experiment control frame was calculated and plotted in Figure 3. From the figure, it is evident that the DCR of the SPAD array is not consistent and varies with each control set taken. However, what is consistent is the DCR pattern for each bias voltage. Certain environmental factors like temperature and surrounding ambient light were not controllable.

As the control DCRs vary between experiments and the low expected total incident exposure intensity of the pion beam, the calculated DCR noise floor for each experiment must be calculated using the respective control data set taken before each experiment for each respective bias voltage. Therefore, a total of four DCRs are calculated for each experiment, with the calculated mean and standard deviation for each pixel.

Using the cumulative DCR noise floors for each experiment and each respective bias voltage, a cumulative HDCR list of pixels for the MINI3D SPAD image sensor used was acquired and a total of 2892 pixels were flagged as HDCR pixels and removed for any data analysis. This equates to approximately 18.8% of all pixels, which is below the expected 20%, which display a significant variation in the distribution of DCR [6]. A heatmap of flagged HDCR is shown in Figure 4, with HDCR pixels indicated in black. For each frame taken during each experimental irradiation, all HDCR are removed from the analysis, and for the remaining pixels, the average counts per pixel (Hz) for each frame are calculated.

For experiment 1, the average counts/pixel vs. frame for before, during and after irradiation (separated by vertical dotted lines) are combined for each SPAD bias voltage and shown in Figure 5. From the figure, during pion beam irradiation, the average counts per pixel increase for all bias voltages, though fluctuating and with a gradual increase in base average counts. It is also observed that once irradiation is finished, the average counts at each bias voltage remain high with reduced variation. The analysis is repeated for experiments 2 through to 4 and shown in Figure 6, Figure 7 and Figure 8, respectively. For experiment 2 and 3, it is evident that the DCR returns to its original state, therefore indicating that the large increase in DCR after irradiation is impermanent. However, for experiment 4, the variation of the DCR for all bias voltages is significantly larger, indicating potential radiation damage to the SPAD image sensor after irradiation.

For pion experiment 2, the average counts per pixel vs. frames are shown in Figure 6, where there is no apparent increase or change in average counts before, during or after irradiation, except for a gradual increase in base average counts. Hence, as experiment 2 was conducted 6 days after experiment 1 and beam parameters may have changed, e.g., beam spot size, it was assumed the image sensor was misaligned and not directly within the pion beam. Therefore, for experiment 3, the 3D motorized stage on the telescope was used to move the SPAD image sensor in increments of 1 mm until a change in average counts was observed. As can be seen in Figure 7 at frame 40,001, there is a distinct increase in average counts across all bias voltages after the image sensor position was changed during irradiation to within the beam line. 

For experiment 3 during irradiation, the average counts have a minor increase in variation when compared to average counts before irradiation; however, this does not resemble the response measured in experiment 1 in Figure 5, which shows a much larger variation during irradiation. Additionally, comparing average counts for experiments 1 and 3 after irradiation, it is shown that for experiment 3, the average counts decrease significantly when the pion beam is off but after experiment 1, the same behavior is not observed.

For pion beam experiment 4, the average counts vs. frame for before, during and after irradiation are shown in Figure 8. Initially, the image sensor required the motorized stage to align the sensor within the pion beam, indicated with the constant recorded operation for the SPAD bias voltage of 13 V, until a significant change was observed. During irradiation, the average counts increase; however, the increase in average counts is not consistent, but incrementally increases for each consecutive cycle until the beam has ended. This increment is assumed to be a pre-set by the main user of the beam. Furthermore, unlike experiment 1, both experiment 3 and 4 have slight increases in average count variation during irradiation when compared to average counts before irradiation. However, for experiment 4, the average count variation after irradiation is significantly larger when compared to before irradiation and permanent, indicating radiation damage to the SPAD image sensor.

With the average DCR calculated for every pixel at each respective bias voltage, for each experiment, the total number of pions detected for every frame during irradiation was calculated. Figure 9, Figure 10 and Figure 11 show the distribution of 120 GeV pion counts detected/pixel during irradiation and Table A1, Table A2 and Table A3, at the end of the article, provide the complete statistical results for experiments 1, 3 and 4, respectively.

With Figure 9, Figure 10 and Figure 11 and Table A1, Table A2 and Table A3, it is evident that the measured pion counts/frame are significantly higher than the calculated value of 75 kHz and optimal value of 1.554 MHz. Furthermore, with increased bias voltage, there is a distinct increase in counts. Table 2 provides the average pion counts/frame ($\bar{x}$) and standard deviation (σ) for each pion experiment vs. SPAD bias voltage and they are plotted in Figure 12. From the figure, it is evident that the increase in average counts is linear for an increase in SPAD bias voltage. The standard deviation for experiment 4 is significantly larger as a result of the gradual increase in pion beam intensity during irradiation.

## 6. Conclusions

The presented results confirm the detection of incident high-energy pions using SPADs with the SPS accelerator at CERN. The results confirm that SPADs can be used to detect high-energy ionizing particles, which complements the variety of ionizing radiation that can be detected with CMOS SPADs. 

For the pion detection experiments, the measured average counts per frame are significantly larger than the calculated range of 75 kHz to 1.55 MHz, with a minimum measured average counts/frame of 8.3 MHz at an SPAD bias voltage of 12.5 V and a maximum of 432.1 MHz at 15 V during pion irradiation. Furthermore, a positive linear correlation between the average counts and increasing SPAD bias voltage is observed.

For SPADs, an increase in reverse bias voltage results in a proportional increase in detection probability; therefore, the increase in detected counts during pion irradiation may be attributed to increased impact ionization, electromagnetic cascade, afterpulsing and crosstalk. Additionally, electron–hole pair generation in the silicon bulk around the depletion region may occur, causing secondary free carriers to drift into the depletion region, resulting in additional spurious counts. Regardless, the exact intensity, trajectory and total quantity of incident pions per frame is unknown; therefore, further examination is required to investigate the particle detection efficiency of the device as well as calculation of the TID. Furthermore, after pion irradiation, the SPAD image sensor DCR was permanently altered as a result of ionizing radiation damage; thus, further investigation into the radiation hardness of the SPAD image sensor is required. However, it must be noted that the SPAD was exposed to approximately 13 days of pion spills, which equates to approximately 8 h of cumulative pion irradiation, lasting significantly longer than anticipated.

## Figures and Tables

**Figure 1 sensors-23-08759-f001:**
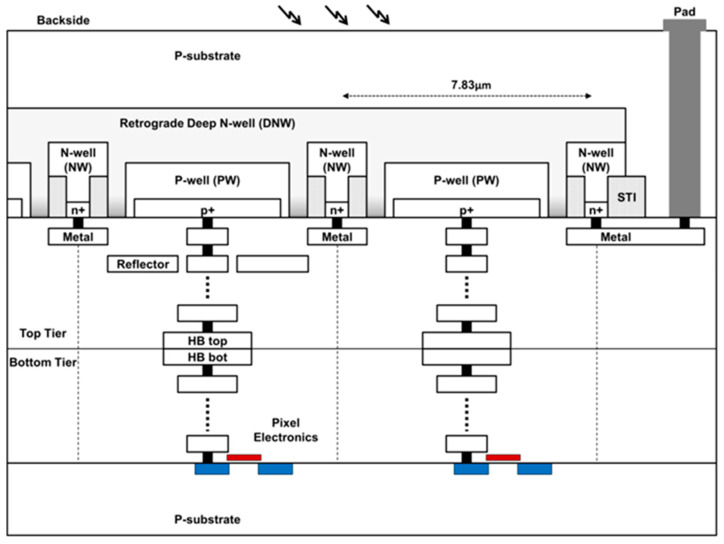
Illustrated cross-section of the MINI3D BSI 3D-stacked pixel layout from [6] (Section 5.1.3). The arrows indicate where photons incident upon the backside of the image sensor enter. The highlighted red and blue blocks indicate the electronic components (transistors) within the bottom tier, where the anode of each SPAD pixel is connected through the via stack from the top to the bottom tier.

**Figure 2 sensors-23-08759-f002:**
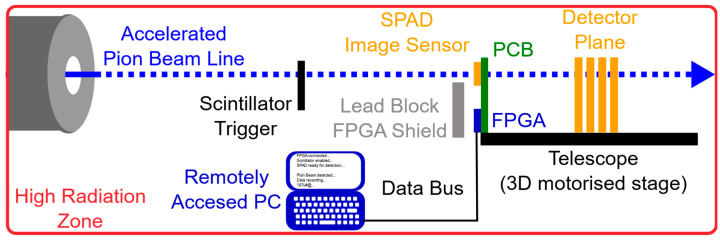
Schematic diagram of the experimental setup, with MINI3D BSI CMOS SPAD image sensor mounted to telescope for pion beam irradiation.

**Figure 3 sensors-23-08759-f003:**
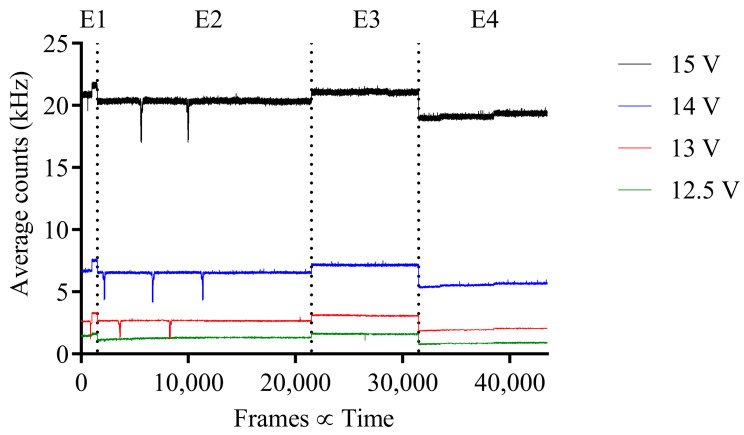
MINI3D DCR (kHz) per frame for each control data set taken before irradiation for each experiment, 1 to 4.

**Figure 4 sensors-23-08759-f004:**
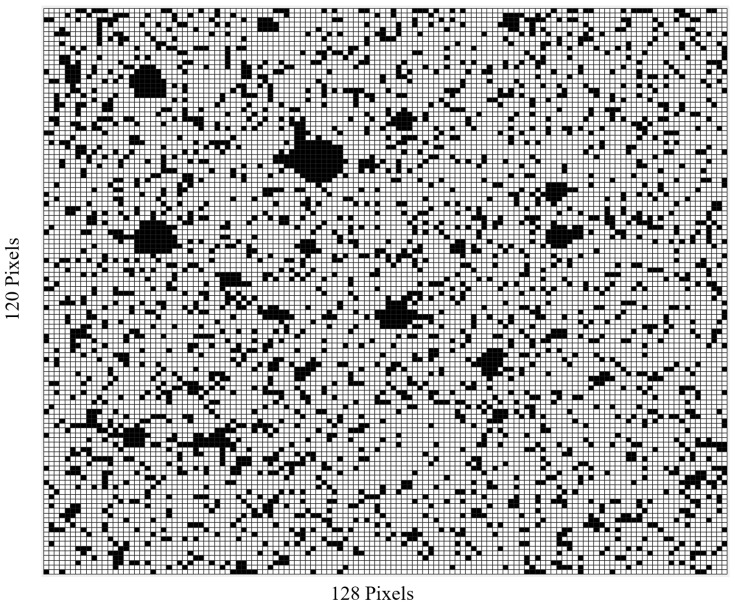
HDCR pixel heatmap for MINI3D CMOS SPAD image sensor used for pion beam experiment. HDCR pixels indicated in black.

**Figure 5 sensors-23-08759-f005:**
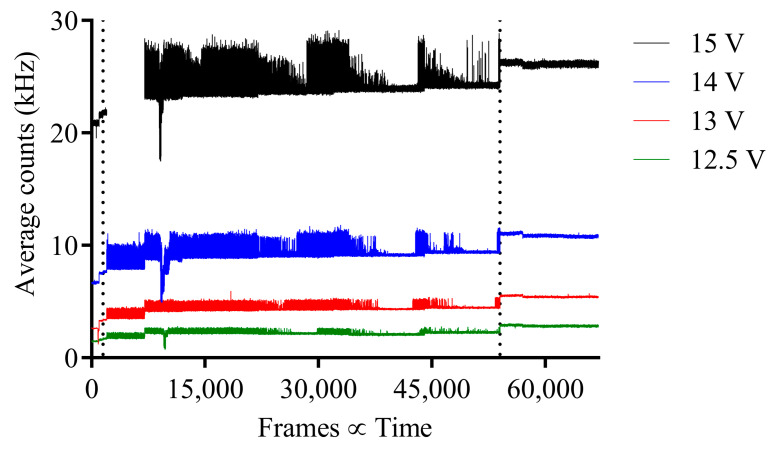
Pion experiment 1; average counts per pixel vs. frame for before, during and after pion beam irradiation for each respective SPAD bias voltage with 120 GeV pion beam. Before, during and after irradiation separated by vertical dotted lines.

**Figure 6 sensors-23-08759-f006:**
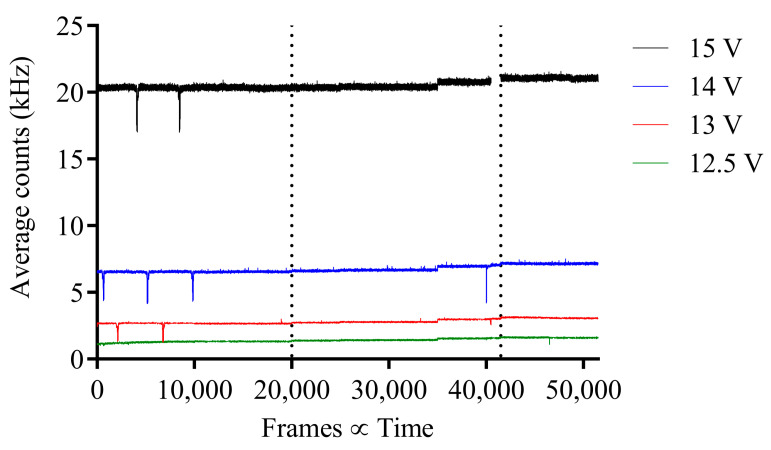
Pion experiment 2; average counts per pixel vs. frame for before, during and after pion beam irradiation for each respective SPAD bias voltage with 120 GeV pion beam. Before, during and after irradiation separated by vertical dotted lines.

**Figure 7 sensors-23-08759-f007:**
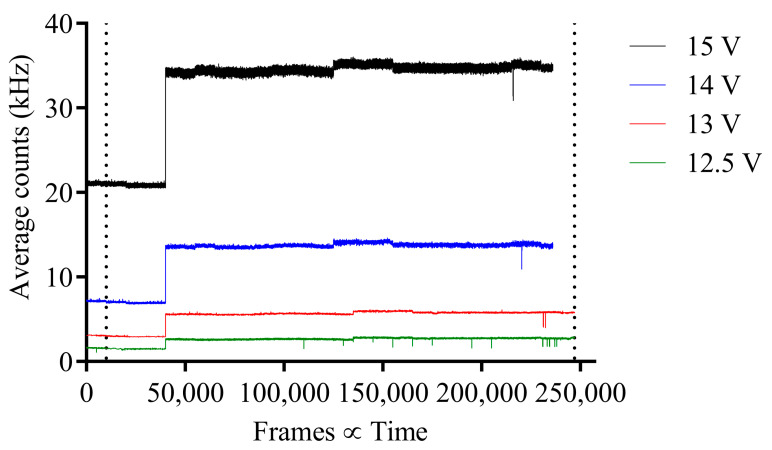
Pion experiment 3; average counts per pixel vs. frame for before, during and after pion beam irradiation for each respective SPAD bias voltage with 120 GeV pion beam. Before, during and after irradiation separated by vertical dotted lines.

**Figure 8 sensors-23-08759-f008:**
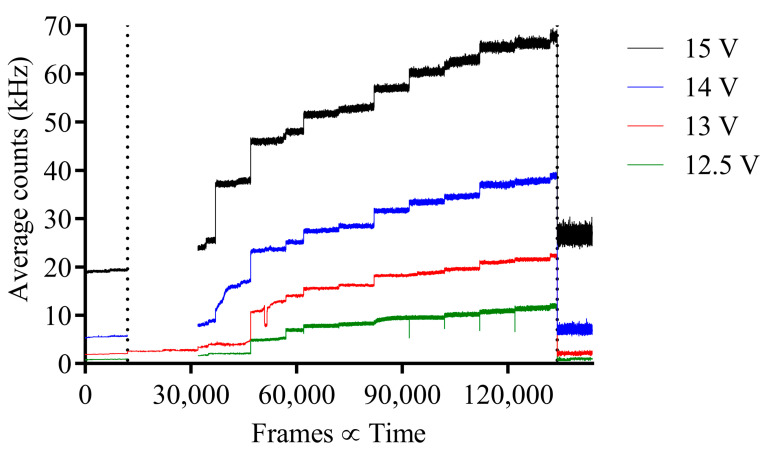
Pion experiment 4; average counts per pixel vs. frame for before and during pion beam irradiation for each respective SPAD bias voltage with 120 GeV pion beam. Before and during irradiation separated by a vertical dotted line.

**Figure 9 sensors-23-08759-f009:**
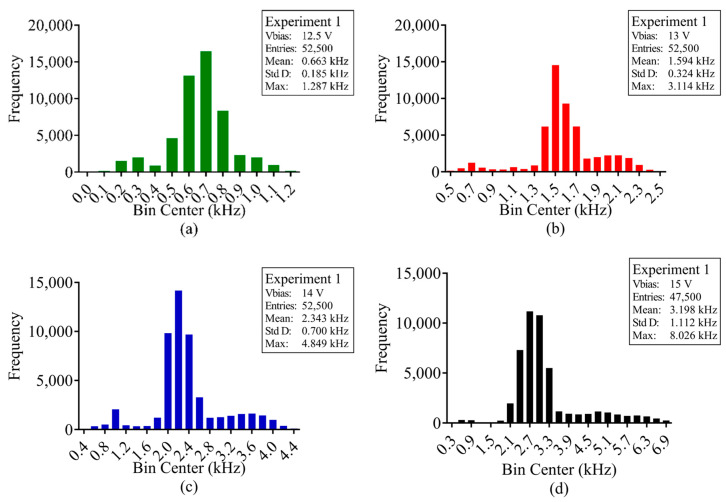
Pion beam experiment 1; frame distribution of average counts per pixel (kHz) during pion beam irradiation for SPAD bias voltage (**a**) 12.5 V, (**b**) 13 V, (**c**) 14 V and (**d**) 15 V.

**Figure 10 sensors-23-08759-f010:**
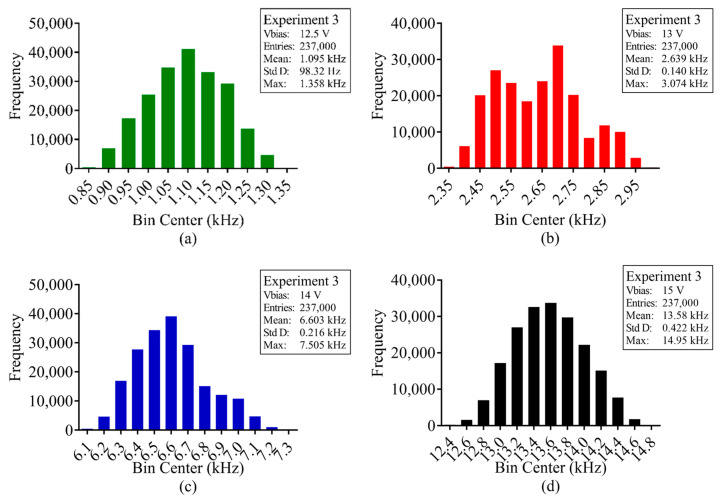
Pion beam experiment 3; frame distribution of average counts per pixel (kHz) during pion beam irradiation for SPAD bias voltage (**a**) 12.5 V, (**b**) 13 V, (**c**) 14 V and (**d**) 15 V.

**Figure 11 sensors-23-08759-f011:**
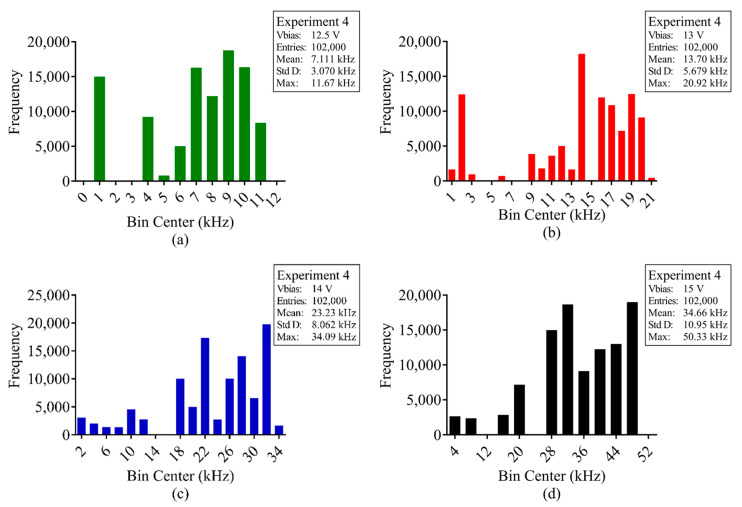
Pion beam experiment 4; frame distribution of average counts per pixel (kHz) during pion beam irradiation for SPAD bias voltage (**a**) 12.5 V, (**b**) 13 V, (**c**) 14 V and (**d**) 15 V.

**Figure 12 sensors-23-08759-f012:**
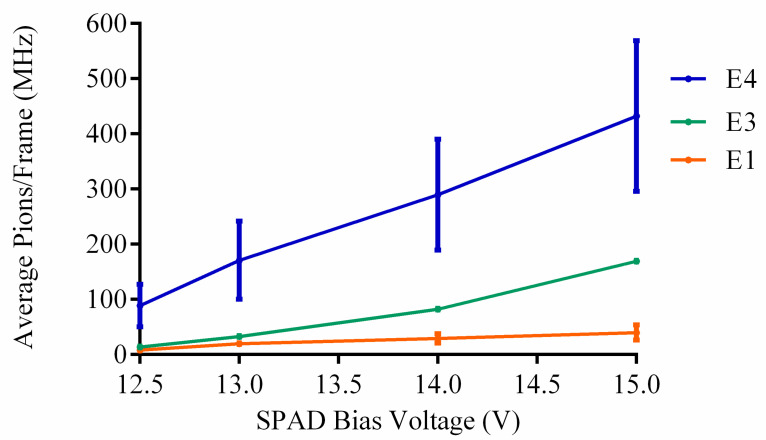
Average pion counts/frame including standard deviation as a function of SPAD bias voltage for each respective pion experiment.

**Table 1 sensors-23-08759-t001:** Pion beam experiments’ total frames taken before, during and after irradiation, with respective start and end dates in date month year format.

	Experiment 1			Experiment 2		
	Control Before	Irradiation	Control After	Control Before	Irradiation	Control After
Frames	1500	52,500	13,000	20,000	21,500	10,000
Date (start–end)	24 October 2018	24–26 October 2018	26 October 2018	1 November 2018	1–2 November 2018	2 November 2018
	**Experiment 3**			**Experiment 4**		
	Control Before	Irradiation	Control After	Control Before	Irradiation	Control After
Frames	10,000	237,000	10,000	12,000	102,000	10,000
Date (start–end)	2 November 2018	2–7 November 2018	7 November 2018	9 November 2018	9–12 November 2018	12 November 2018

**Table 2 sensors-23-08759-t002:** Pion average ($\bar{x}$) counts/frame and standard deviation (σ) as a function of SPAD bias voltage for experiments 1, 3 and 4.

	Average Pion Counts/Frame ^a^ and Standard Deviation (MHz)					
Bias Voltage	Experiment 1		Experiment 3		Experiment 4	
	$\bar{x}$	*σ*	$\bar{x}$	*σ*	$\bar{x}$	*σ*
12.5 V	8.3	2.3	13.7	1.2	88.7	38.3
13 V	19.9	4.0	32.9	1.7	170.8	70.8
14 V	29.2	8.7	82.3	2.7	289.7	100.5
15 V	39.9	13.9	169.4	5.3	432.1	136.5

^a^ 12,468 measured pixels.

## Data Availability

Unavailable at time of submission.

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
