# Peer review of "Pion Detection Using Single Photon Avalanche Diodes"

_sensors, 2023, doi:10.3390/s23218759_

Round 1

Reviewer 1 Report

The authors used a CMOS compatible SPAD array for pion detection, and it was confirmed that the pions can be detected by the device. However, the lack of control and records of the conditions in the  experiments makes it hard to evaluate the value of the application of SPAD array in pion detection.

1. In the manuscript, all the references to figures are lost and are shown as "Error! Reference source not found.", which leads to difficulties in understanding of the contents.

2. At the end of page 8, the authors mentioned that "Certain environmental factors like temperature and surrounding ambient light were not controllable." However, it is possible to control the working temperature of the sensor, or at least a temperature-performance function could have been established before the experiments. And the record of temperature during the experiments might support the data analysis. As for surround ambient light, it could greatly affect the count rate when presents, and I believe it can be well blocked through the application of filters. Thus, in my opinion, efforts could be made to avoid the effects of ambient light and temperature fluctuance.

3. The dose of pions were not well controlled and/or not well recorded, and hence the capability of pion detection for the SPAD array remains unknown to the authors. E.g., standard detection methods such as the use of scintillators with PMTs could have been adopted in addition to the SPAD array in the experiments as a comparison.

4. The measured results were much larger than the calculated values, but the authors did not provide sufficient analysis.

Author Response

Please find word file attached with authors reply. Thank you

Reviewer 2 Report

Original paper on the first use of a SPAD sensor for pion detection.

If the second part is dedicated to experimental results that are well described, the first part sets the context and is quite unclear. No comparison is given with other technologies. So the choice of a SPAD sensor is not obvious. The choice of the pixel dimensions is not explained either.

Moreover, the words "ionizing radiations" is often used wheras it gathers different particles that can't be detected the same way. For example, X-ray detection is mentionned as a potential application for silicon SPAD sensors but this assumption is highly questionable.

For these reasons, I would recommand the authors to revise their first part to better describe the real interest of the SPAD sensor for their application.

Some comments or questions are embedded in the pdf file.

Author Response

(The authors gave the same response as above.)

Reviewer 3 Report

The paper presents interesting results on registration of pions at SPS in CERN using a SPAD array.   It could be published,  if  significantly shortened and with  focusing on  what is written  in the title and abstract.

1. The first 3 sections repeat published in many papers general info on SPADs.  Some of the graphs are confusing. For example, Fig.1 is , probably, not the I-V, but "gain as a function of the applied voltage" (log scale, like in ref [5] ).

It is not clear, why the discussion on the guard ring types  in SPAD pixels has relation to using a concrete SPAD array  for registration of pions . ..

I  suggest to start the paper with the description of the used BSI imager (section 4) and exclude (or radically shorten) all three first paragraphs.

2. Reference to SPAD sensor , as originating from the university lab, looks strange. We  are discussing the BSI SPAD imager  fabricated in STMicroelectronics, as stated in [5] ...Correct?

4. The explanation of signal increase in time by electrons/holes generated outside the depletion region of SPAD is not clear. Using other types of radiation  (e.g. X-rays) could be , probably, useful   to  investigate   the  observed  signal evolution in time.

5. References  to  figures and literature  are absent in many places  of  the paper.  This  strongly complicates the review.

Author Response

(The authors gave the same response as above.)

Round 2

Reviewer 1 Report

Since all comments were addressed, I have no further suggestions.

Author Response

No more comments. 

We would like to thank the reviewer for taking the necessary time and effort to review the manuscript. We sincerely appreciate all your valuable comments and suggestions, which helped us in improving the quality of the manuscript.

Reviewer 2 Report

Dear authors,

I don't think SPAD are relevant for X-ray detection as far-more-efficient sensors are available. A SPAD will detect, depending on its thickness and on the photons energy, less than 1 out of 100 X-ray photons. That is a very poor detection.

What I suggest is that you remove all X-ray mentions in your text, even in the introduction. It will make your message clearer and X-ray experts will not be confused about an assertion only supported by your own previous papers. 

Some other comments are in the file attached.

Best Regards

Author Response

Reviewer             2

Journal                 Sensors (ISSN 1424-8220)

Manuscript ID    sensors-2467253

Type                      Article

Title                       Pion Detection using Single Photon Avalanche Diodes

Authors                Anthony Frederick Bulling * , Ian Underwood

Section                 Optical Sensors

Special Issue      Optical Sensors Technology and Applications: Volume II

Abstract: We present the first reported use of a CMOS compatible Single Photon Avalanche Diode (SPAD) array for the detection of high energy charged particles, specifically pions, using the Super Proton Synchrotron (SPS) at CERN, the European Organization for Nuclear Research.

Comments and Suggestions for Authors

Dear authors,

I don't think SPAD are relevant for X-ray detection as far-more-efficient sensors are available. A SPAD will detect, depending on its thickness and on the photons energy, less than 1 out of 100 X-ray photons. That is a very poor detection.

What I suggest is that you remove all X-ray mentions in your text, even in the introduction. It will make your message clearer and X-ray experts will not be confused about an assertion only supported by your own previous papers. 

Some other comments are in the file attached.

Best Regards

Author’s reply to Review Report:

We would like to thank the reviewer for taking the necessary time and effort to review the manuscript. We sincerely appreciate all your valuable comments and suggestions, which helped us in improving the quality of the manuscript.

Regarding the comments, the Author’s agree with the viewers comments, and will omit the mention of X-Rays in the paper.

Replies to comments in attached pdf, are addressed to the reviewer in the same pdf.

Best wishes.

Reviewer 3 Report

My comments were addressed in the revised manuscript.

Author Response

(The authors gave the same response as above.)
